# Contribution of ICOH to Address the Different OSH Needs among Countries: Results of a Survey

**DOI:** 10.3390/ijerph18094665

**Published:** 2021-04-27

**Authors:** Bruna Maria Rondinone, Antonio Valenti, Valeria Boccuni, Erika Cannone, Pierluca Dionisi, Diana Gagliardi, Fabio Boccuni, Sergio Iavicoli

**Affiliations:** 1Department of Occupational and Environmental Medicine, Epidemiology and Hygiene, Italian Workers’ Compensation Authority (INAIL), Via Fontana Candida 1, Monte Porzio Catone, 00078 Rome, Italy; b.rondinone@inail.it (B.M.R.); v.boccuni@inail.it (V.B.); p.dionisi@inail.it (P.D.); d.gagliardi@inail.it (D.G.); f.boccuni@inail.it (F.B.); s.iavicoli@inail.it (S.I.); 2International Commission on Occupational Health (ICOH), Via Fontana Candida 1, Monte Porzio Catone, 00078 Rome, Italy; icoh@inail.it

**Keywords:** ICOH, OSH needs, occupational health, survey, multiple correspondence analysis (MCA)

## Abstract

The aim of this study is to map the coverage of occupational safety and health (OSH) rules and provisions and their enforcement at a country level worldwide. Members’ participation in the International Commission on Occupational Health (ICOH) activities was also investigated. We used a questionnaire-based survey to collect data. An online questionnaire was administered from February 14 to March 18, 2018 to all ICOH members for the triennium 2015 to 2017 (*n* = 1929). We received 384 completed questionnaires from 79 countries, with a 20% response rate. To synthesize information about the coverage of OSH rules and provisions and their level of enforcement, a synthetic coverage index was calculated and combined with country, gross domestic product (GDP) per capita and the human development index (HDI). We used multiple correspondence analysis (MCA) to analyze the members’ participation in ICOH activities. More than 90.0% of the sample declared that in their own country there is a set of rules and provisions regulating OSH in the workplace, and training procedures and tools to improve workers’ awareness. However, these rules and training procedures are mainly “partially” enforced and utilized (39.0% and 45.4%). There was no statistically significant association between country and GDP per capita and the synthetic coverage index, whilst controlling for HDI. The level of engagement in ICOH activities is higher in senior members (aged 65 years or older), coming from high-income countries, having held a position within ICOH, with a higher level of education and a researcher position. An integrated and multidisciplinary approach, which includes research, education and training, is needed to address OSH issues and their impact both at global and country level.

## 1. Introduction

Despite global efforts to address occupational safety and health (OSH) concerns, it is estimated that each year 2.78 million workers die from work-related injury and disease and 313 million workers suffer from non-fatal work-related injury and illness. It is estimated that lost workdays globally represent almost four percent of the world’s gross domestic product (GDP), and in some countries, this rises to six percent or more [1,2]. In addition, only 15% of workers worldwide have access to specialized occupational health services carrying out the prevention of occupational risks, health surveillance, training in safe working methods, first aid and advising employers in occupational health and safety [3].

Over the last century, the United Nations (UN), World Health Organization (WHO) and International Labour Organization (ILO) have identified OSH as one of their key priorities within the general framework of sustainable development [4]. In September 2015, the UN General Assembly adopted the 2030 Agenda for Sustainable Development including workers’ health and decent jobs among its goals [5]. The inclusion of this important issue in the UN resolution marks a continuity with the WHO Global Plan of Action on Workers’ Health (2008 to 2017), which promotes the protection of health at the workplace through the adoption of regulations and a basic set of occupational health standards. Consistently with WHO and UN policies, ILO calls on member states and social partners to strengthen OSH systems as prescribed by Convention 187 (Promotional Framework for Occupational Safety and Health) and Convention 155 (Occupational Safety and Health). The UN political declaration on universal health coverage, adopted by the UN General Assembly on October 10, 2019, strongly recommits to achieve universal health coverage by 2030, with a view to scaling up the global efforts to promote a healthier world for all. The efforts include raising awareness, sharing knowledge and best practices, strengthening health information systems and the engagement of all stakeholders, including OSH experts’ community [6].

The joint commitment of these supranational institutions highlighted this important issue. However, a considerable burden remains for governments, employers, workers and other stakeholders in building present and future generations of safe and healthy workers, especially considering the emerging OSH risks due to the introduction of new technologies, substances and work processes, together with changes in the labor market, and with new forms of employment and work organization [7].

In this regard, the development and dissemination of information and knowledge that may meet the needs of governments, employers and workers are a prerequisite for identifying priorities, adopting relevant strategies and implementing national OSH programs. ILO considers international and regional networks pivotal to the effective development and exchange of OSH knowledge and data and calls for the contribution of non-governmental organizations to achieve this objective.

In this perspective, the International Commission on Occupational Health (ICOH), the oldest and leading global professional organization in the OSH field [8], has called for the organization of occupational health services, either basic or comprehensive, for all working people, promoting the principle of universality [9,10,11]. A recent study conducted by ICOH in 2017 [12] involving the network of ICOH national secretaries showed that the majority of the countries had drawn up policies, strategies and programs for occupational health services. However, the study also revealed a wide gap in the implementation of policies into practice, leaving the majority of workers without access to occupational health services.

Based on the interesting results that emerged and in view of further complementing the data on occupational health service coverage and comparing them with data on the implementation of OSH policies worldwide, ICOH developed the present study, which was addressed, for the first time in ICOH history, to the entire active membership.

The aim of the present study is to map the coverage of OSH rules and provisions, their level of enforcement in each participating country, and the members’ participation in the ICOH activities.

In the following sections, we will present an overview of the relevant literature and will describe the research methods and statistical analysis. We will then discuss the results, the limitations and the practical implications of this study.

## 2. Literature Review

### 2.1. Occupational Safety and Health

OSH is generally defined as the science of the anticipation, recognition, evaluation and control of hazards arising in or from the workplace that could impair the health and well-being of workers, taking into account also the possible impact on the surrounding communities and the general environment [13].

As defined by the WHO, health is “a state of complete physical, mental and social well-being and not merely the absence of disease or infirmity” [14]. Occupational health deals with all aspects of health and safety in the workplace and has a strong focus on the primary prevention of hazards [15].

The protection of the worker against sickness, disease and injury arising out of employment is not only a labor right, but a fundamental human right and is one of the main objectives of the ILO as stated in its Constitution [16]. Over the years, measures and strategies designed to prevent, control, reduce or eliminate occupational hazards and risks have been developed and have evolved gradually and continuously in response to social, political, technological and economic changes [13]. Occupational health has therefore gradually developed from a monodisciplinary risk-oriented activity to a multidisciplinary and comprehensive approach [17].

### 2.2. International Commission on Occupational Health

ICOH is the oldest scientific association in the field of OSH, founded in 1906. Today, ICOH counts more than 2000 members in over 100 countries. ICOH has promoted occupational health for decades, in line and in collaboration with the UN, ILO, WHO, as well as with other non-governmental organizations and professional associations. In this way, ICOH has become a forum for the exchange of information and knowledge among OSH experts and professionals at an international level [8].

Thanks to the activities of its 37 scientific committees, ICOH covers research, training, information, and the introduction of good practices of occupational health services. ICOH activities also include the monitoring and follow-up, at a global level, on the development of occupational health services, which represent the principal infrastructure for workers’ health in the world of work.

ICOH has called for the organization of occupational health services, either basic or comprehensive, for all working people, promoting the principle of universality, through a number of declarations and statements [9,10,11]. In addition, ICOH has carried out two surveys on occupational health services, in collaboration with individual researchers, research groups or scientific committees [12,18,19,20].

The first ICOH survey on occupational health services was carried out in 2010 to 2011. It was conducted among the ICOH national secretaries to explore the status of occupational health services at a national level. The collected data were used for a global estimate [18]. The second ICOH survey on occupational health services focused on their normative basis, structures, resources, functions, service provision systems, coverage and future development needs. ICOH national secretaries served again as key informants. This allowed to highlight changes occurring after the previous survey, which was conducted five years earlier [12].

### 2.3. OSH Rules and Provisions

Occupational health can be defined through the state of its legal framework, the capacity of the institutions responsible for the enforcement of OSH rules and provisions, and the actual implementation and enforcement of OSH rules and provisions. The level of awareness about OSH issues is an important determinant of the status of OSH in a country.

## 3. Research Methods

### 3.1. Research Approach

This was a cross-sectional study conducted through an online questionnaire. The questionnaire survey is a research method involving the use of questionnaires to collect data directly from persons involved in the research through a set of questions organized in a particular order and intended to capture responses in a standardized manner. This has become one of the most frequently used methods for quantitative research. It allows obtaining information about a given phenomenon by formulating questions that reflect the thoughts, opinions and perceptions of a group of individuals [21].

One of the most recent types of questionnaire survey is an online or web survey that can be administered by forwarding a web link by email. These systems offer several benefits such as low cost and immediate availability of the results in an online database. The management of the surveying campaign is also very simple, and allows a real time check of responses, scheduling of reminders, etc.

This type of questionnaire survey was selected for the present study because of the possibility to reach the entire population at a low cost.

### 3.2. Instrument Development

The questionnaire sections included contact information (i), ICOH contribution to OSH at a national and international level (ii), scientific committees (iii), national secretaries (iv), international congresses (v), ICOH communication tools (vi), and general secretariat management and operating activities (vii). A preliminary pilot test was conducted to collect feedback on the clarity of questions and response options, sequence, flow and accessibility of the online platform. The pilot study involved 16 senior ICOH members such as the officers, selected board members and national secretaries. Suggestions and observations gathered were considered to develop the final version of the questionnaire. Following an informative email describing the study, all ICOH members received an electronic invitation generated by the SurveyMonkey system with a link to the online questionnaire. The questionnaire was administered from February 14 to March 18, 2018, with one reminder in order to increase the response rate. The questionnaire was circulated in English, which is the official language of ICOH.

### 3.3. Sampling and Data Collection

The online questionnaire was administered to all 1929 ICOH members in good standing for the triennium 2015 to 2017 through the dedicated web-based platform SurveyMonkey. No sampling was performed because the questionnaire was addressed to the whole population of ICOH members.

### 3.4. OSH Rules and Training Procedures: Level of Enforcement and Utilization

The following questions were used to construct a synthetic coverage index: “Do you have in your country a set of rules and provisions, which regulate occupational safety and health in the workplace?” (Yes, No, I don’t know) and “Are there in your country training procedures and tools to improve workers’ awareness and knowledge on the protection of health and safety in the workplace?” (Yes, No, I don’t know). In the cases of a positive reply to the above questions, the respective levels of enforcement and utilization were also considered, using a five-point scale (ranging from 1 as the minimum level, to 5 as the maximum level). Data referred to the four variables were normalized using the minimum and maximum value formula yi = (xi-xmax)/(xmax-xmin). Then, the weighted average was calculated based on the number of responses given by each respondent to the four questions. “Don’t know” answers were considered as missing values. The index thus obtained (continuous variable ranging from 0 as the minimum coverage level to 1 as the maximum coverage level) was combined with the country, GDP per capita (categorical variables) and 2016 human development index (HDI) (continuous variable). The HDI is a summary measure of average achievement in key dimensions of human development: a long and healthy life, being knowledgeable and having a decent standard of living (http://hdr.undp.org/en/data#).

### 3.5. Members’ Participation in ICOH Activities

The members’ participation in ICOH activities was evaluated, taking into simultaneous consideration a list of variables classified in two groups. The first one includes all demographic variables, such as gender, age (25 to 44, 45 to 64, 65 and older), country (Africa, America, Asia, Europe, Oceania), GDP per capita in USD (low, medium, high), education (Bachelor’s degree or lower title, Master’s degree, PhD, other title), main activity (academician, researcher, practitioner, other activity), and languages spoken other than mother tongue (English, other languages).

The second group contains variables regarding the participation in ICOH activities. The variables included are: year of joining ICOH (before 2010; 2011 and after); Do you have a position within ICOH? (Y/N); Do you belong to other scientific organizations? (Y/N); Are you a member of one or more ICOH scientific committees? (Y/N); Have you ever attended an ICOH international congress? (Y/N); How many ICOH congresses have you attended so far? (1–3; 4–7; More than 7); Have you ever submitted a scientific contribution on the occasion of an ICOH congress? (Y/N); Have you ever attended the General Assembly? (Y/N); Have you ever voted to select ICOH congress venue? (Y/N); Have you ever voted to elect the ICOH officers and board members? (Y/N); How often do you visit the ICOH website? (at least on a monthly basis, rarely or never, only when I need specific information); Do you consult the ICOH newsletter? (Y/N); Do you use social media for your professional life? (Y/N); Do you have the ICOH app downloaded on your device? (Y/N).

### 3.6. Statistical Analysis

A two-way analysis of covariance (ANCOVA) was conducted to examine the effect of country and GDP per capita on the synthetic coverage index, after controlling for HDI.

Before the ANCOVA was run, all assumptions for this test were verified: linear relationship between dependent variable and covariate for each combination of the groups of the two independent variables, homogeneity of regression slopes, homoscedasticity, homogeneity of variance (Levene’s test of equality of error variance), normality (Shapiro–Wilks).

To analyze the participation of members in ICOH activities, multiple correspondence analysis (MCA) was performed. MCA is a multivariate exploratory analysis for visualizing large datasets of categorical variables. Its graphical visualization provides a structural organization for the variables and categories in a dimensional space that is useful for identifying patterns in the data and associations between the investigated parameters [22,23]. In this analysis, two types of variables are considered: active variables and supplementary variables. Active variables are those that contribute directly to the formation of the low-dimensional space and to the definition of the factors considered. Supplementary variables are those that are not useful for the determination of the principal dimensions but are useful to better describe the dataset and the latent factors. In other terms, active variables provide a subjective description of the units of analysis, while supplementary variables assess different groups of observation and belong to a specific category. In this study, active variables are those regarding the participation in ICOH activities and supplementary variables are demographic variables. Two variable categories are directly associated (direct relationship) if they have both high coordinate positions and are in the same quadrant of the MCA plot. Two variable categories are inversely associated if they both have high coordinate positions but opposite signs.

The answers to questions concerning the sources consulted for professional training in OSH and possible difficulties found in consulting such sources were also analyzed and then combined with socio-demographic variables.

### 3.7. Demographics

The questionnaire was sent to all 1929 ICOH members in good standing for the triennium 2015 to 2017. A total of 384 members from 79 countries completed the questionnaire. The response rate was 20%. Demographic and professional details of respondents are described in Table 1. Most of the respondents were male (58.1%), aged between 45 and 64 years (53.6%), holding a PhD degree (45.8%), coming from Europe (34.9%) and from high-income countries (48.4%). The respondents were also asked to indicate which languages they could speak other than their mother tongue. English came first (51.2%), followed by French (13.1%). As for the profession, most of the respondents were physicians (57.0%). A total of 40.9% of the respondents were practitioners. A total of 32.8% of the respondents worked for academia/university. A total of 52.2% of the respondents joined ICOH in 2011 or after.

## 4. Results

### 4.1. OSH Rules and Training Procedures: Level of Enforcement and Utilization

Respondents were asked to indicate the existence in their countries of (1) a set of rules and provisions regulating occupational safety and health in the workplace, and (2) training procedures and tools to improve workers’ awareness and knowledge of health and safety protection in the workplace. More than 90.0% of the sample answered in the affirmative, even though different percentages of missing values and respondents who were unable to answer were also registered.

In the cases of a positive reply to these two questions, it was also asked whether the rules and provisions are effectively enforced, and the training procedures are effectively utilized, using a scale ranging from 1 (minimum level) to 5 (maximum level). Table 2 shows the frequency percentage related to the level of enforcement of rules and provisions and to the utilization of the tools. In both cases, the highest percentage of responses is recorded in the level 2, partially enforced (39.0%) and utilized (45.4%).

The results of the ANCOVA, in particular the means, adjusted means, standard deviations (SD) and standard errors (SE), are presented in Table 3.

There was no statistically significant interaction between country and GDP per capita on the synthetic coverage index, whilst controlling for HDI, F (5, 371) = 1.435, *p* = 0.211, pη2 = 0.02. This indicates that the effect of an independent variable is the same for each level of the other independent variable, after controlling for the covariate. There was a statistically significant main effect of the independent variables country: F (4, 371) = 2.642, *p* = 0.033, pη2 = 0.028, and GDP per capita: F (2, 371) = 3.076, *p* = 0.047, pη2 = 0.016, when controlling for HDI (Table 3).

The adjusted means are the predicted group means for the dependent variable when the covariate is set to its average value.

The main effect of country showed a statistically significant difference in adjusted marginal mean of synthetic coverage index: for Africa (0.777), it was higher than America (0.584) *p* = 0.038.

Additionally, 62.5% of the respondents believe that the role of ICOH in developing scientific knowledge and professional skills is very or totally important, while 23.1% believe that it is fairly important and 14.4% partially or not at all important.

Regarding the influence of the ICOH publications on the development of international OSH policies, 47.8% of the respondents indicated that ICOH publications are much or totally significant, 26.2% fairly significant and 25.9% partially or not at all significant.

### 4.2. Members’ Participation in ICOH Activities

MCA revealed a cumulative inertia of 64.2% for the first three dimensions, meaning that the first three dimensions accounted for 64.2% of the variations observed in the sample. The first-dimension inertia was 31.2%, the second was 20.7%, and the third was 12.3%.

In the first dimension (Figure 1, *x*-axis), the youngest members coming from low-income countries (particularly from Africa), with lower education, who did not belong to any scientific committees, never attended ICOH congresses and joined ICOH only recently (left side of the graph) are opposed to the most senior members coming from high income countries (particularly from Oceania), being in the ICOH community for a longer time, having held a position within ICOH and actively contributing to ICOH activities by attending many ICOH congresses, submitting scientific contributions in the occasion of the congresses, participating in the General Assembly and to the voting procedures for the election of ICOH officers, board members and the congress venue (right side of the graph).

In the second dimension (Figure 1, *y*-axis), the highest contribution comes from those members who never voted to elect ICOH officers and board members or to select an ICOH congress venue, never participated in the General Assembly or submitted a scientific contribution to the ICOH congresses, attended one to three ICOH congresses or none so far and do not consult the ICOH newsletter.

In the third dimension (Figure 2, *y*-axis), the highest contributions are associated with those members who, while not contributing actively to the ICOH activities (never participated in the General Assembly, never voted to elect ICOH officers and board members or to select an ICOH congress venue) have nevertheless downloaded the ICOH app on their device, visit the ICOH website on a monthly basis, consult the ICOH newsletter and belong to other scientific organizations (lower side of the graph).

### 4.3. Sources Consulted for Professional Training in OSH

Among the most used sources for professional training in OSH, the peer reviewed journals and websites come first with the same percentage (18.3% of responses, 81.9% of cases), followed by the conference proceedings (14.4% of responses, 64.5% of cases), Table 4.

20.3% reported to find difficulties in consulting such sources due to difficult access (45.3%) and high costs (37.3%). Other reasons, such as excessively technical language, language gap and others, stand at about 17%. Crossing the variable concerning the difficulties in the consultation of the sources with the socio-demographic variables, there are statistically significant associations in relation to the country (*p* = 0.013), class of age (*p* < 0.001) and GDP per capita (*p* < 0.001) variables. Particularly, Europe has the highest percentage of respondents who declared not to have difficulties (89.2%). Among the other countries, such percentages are below 77% approximately. In relation to age, the percentage of those who did not find difficulties in the consultation rises with increasing age up to 70.6% among the people aged 25–44 years, 78.1% for those aged 45–64 years, and 97.2% among those aged 65 years and older. With reference to the GDP, the percentage of respondents without difficulties in the consultation of the sources ranges from 63.6% for the countries with a low GDP, to 74.6% for the countries with middle GDP and to 89.6% for the countries with high GDP. Finally, the HDI is on average higher (*p* < 0.001) among the respondents without difficulties in the consultation of the sources (0.821) compared to those who reported difficulties (0.746).

## 5. Discussion

The future of work and the ability to produce resources for maintaining the economic and social fabric is critically dependent on the health and work ability of workers and thus on occupational health. This becomes particularly true in the phase of transformation that the world of work is currently undergoing, with occupational health being at different stages around the world. While the poorest countries are still facing problems with occupational health service coverage and accessibility, industrialized countries have to deal with diametrically opposed issues, such as active ageing. As for the middle-income countries, they are experiencing a rising awareness of issues related to OSH.

These needs enhanced various efforts in all countries to obtain more intensive OSH policies, including legislation and enforcement; national programs; service infrastructures with the necessary support services, including training, education, statistics and information systems; and research [12].

In line with its mission of fostering the scientific progress, knowledge and development of OSH in all its aspects, ICOH conducted some relevant surveys addressed to its national secretaries, regarded as key informants, with the aim of examining the current status of occupational health services systems at a country level [12,18].

The previous ICOH surveys were addressed to a single key informant for each country, limiting the extent of the information provided. Instead, this survey involved professionals with different competencies in the OSH field (physicians, epidemiologists, nurses, hygienists, psychologists, etc.) and active within different scientific communities (universities, public governmental institutions, etc.), and this is definitely a strong point of the present study. Even though the 20% response rate may seem unsatisfactory, the socio-demographic characteristics of the respondents (gender, age, profession, etc.) are representative of the ICOH membership in a proportional way. In addition, the geographical distribution of the respondents also ensures a wide representation of the different areas.

Consistently with previous surveys, this study also shows that one of the main challenges for OSH is the implementation gap. In 45% of the countries included in the survey, OSH rules and provisions are not at all or only partially enforced, despite almost all countries (98.7%) having OSH national laws and regulations in place. Similarly, even though more than 90% of the respondents reported the existence of training procedures and tools to improve workers’ awareness and knowledge about the protection of health and safety at the workplace, such tools are not at all or only partially used in 49.4% of the cases. The study highlighted the importance and strategic role of ICOH in addressing OSH needs for the development of healthy, safe, innovative and sustainable workplaces.

An overall 74% of the sample consider ICOH publications totally, very and fairly significant for the development of international OSH policies. In addition, more than 85% of the respondents acknowledge as totally, very and fairly important the role of ICOH in developing scientific knowledge and professional skills. On the other hand, even though most respondents indicated peer-reviewed journals and websites as the main sources consulted for professional training, about 20% of the cases complained about persistent difficulties in consulting such sources, mainly due to high costs and difficult access, with a significant relation with country, age, and country GDP.

It is, however, interesting to note that these main critical issues are raised not only by members from low-income countries, but also by those with a more active participation in ICOH (i.e., those participating in congress assemblies, presenting scientific works at the congresses, senior members, coming from high-income countries, in particular Oceania, with a professional profile of researcher).

## 6. Conclusions

ICOH developed the present study with the purpose of further complementing the data on occupational health services coverage arisen from the previous works and comparing them with data on the implementation of OSH policies worldwide. The study investigated many relevant OSH aspects, such as the role played by scientific associations in filling data and knowledge gaps on OSH and contributing to the goals set by universal health coverage. Furthermore, the level of active participation of the OSH scientific community is taken to act as a multiplying factor.

Previous studies addressed a single key informant for each country, limiting the extent of the information provided. On the contrary, the main innovation of this study is represented by the involvement of the entire ICOH active membership. In fact, the participation of professionals with different competencies in the field of OSH (physicians, epidemiologists, nurses, hygienists, psychologists, engineers, etc.) and active within different scientific communities (universities, public governmental institutions, etc.) is definitely a relevant strong point of the present study).

Compared to the sample size (1929 members), the 20% response rate may appear to be a limitation of the study. However, the socio-demographic characteristics of the respondents (gender, age, profession, etc.) and their geographical distribution is sufficiently representative of the ICOH members. The study showed that an integrated and multidisciplinary approach, including research, education and training, is needed to address OSH issues and their impact both globally and nationally. In this perspective, the OSH community plays a pivotal role in identifying the current status and challenges of global occupational health and safety and in disseminating knowledge. These are essential prerequisites to recognize key priorities, develop coherent and relevant strategies, and implement national OSH programs.

## Figures and Tables

**Figure 1 ijerph-18-04665-f001:**
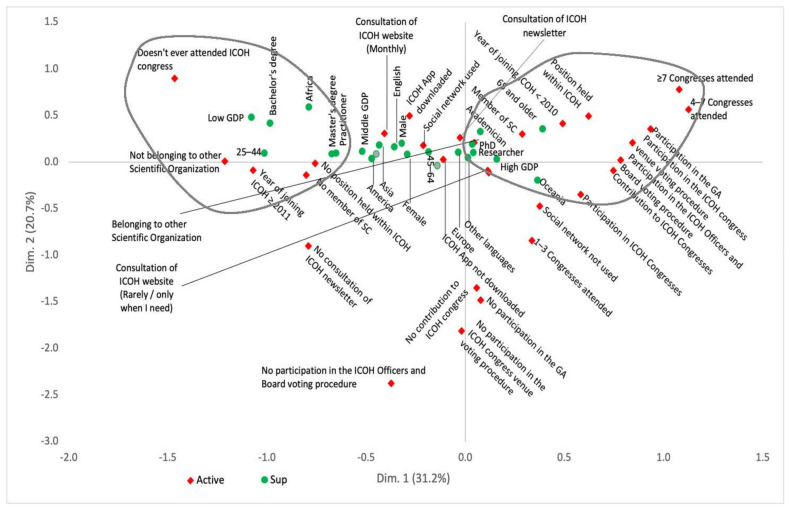
Multiple correspondence analysis map (projections on the first 2 dimensions).

**Figure 2 ijerph-18-04665-f002:**
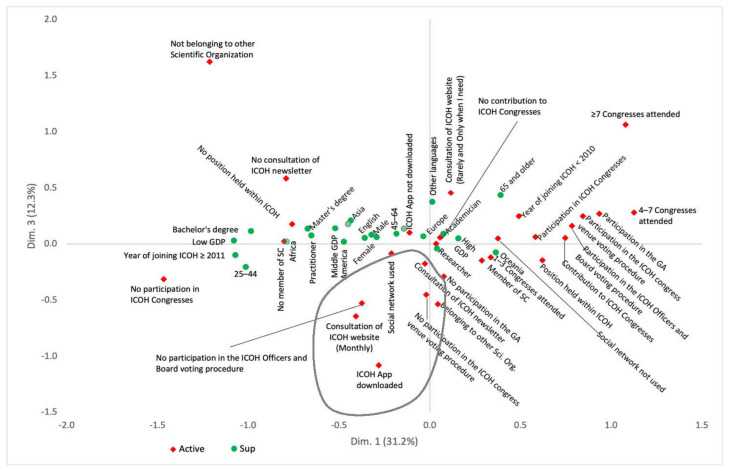
Multiple correspondence analysis map (projections on the first and third dimensions).

**Table 1 ijerph-18-04665-t001:** Description of sample.

Variables	Description	*n* (%)
Gender	Male	223 (58.1%)
	Female	161 (41.9%)
Age group	25–44 yrs	107 (27.9%)
	45–64 yrs	206 (53.6%)
	65 and older	71 (18.5%)
Country	Europe	134 (34.9%)
	America	98 (25.5%)
	Asia	82 (21.4%)
	Africa	50 (13.0%)
	Oceania	20 (5.2%)
Education	High school	7 (1.8%)
	Bachelor’s degree	30 (7.8%)
	Master’s degree	121 (31.5%)
	PhD	176 (45.8%)
	Other	50 (13.0%)
Profession	Physician	219 (57.0%)
	Epidemiologist	34 (8.9%)
	Nurse	25 (6.5%)
	Hygienist	15 (3.9%)
	Engineer	12 (3.1%)
	Psychologist	12 (3.1%)
	Toxicologist	10 (2.6%)
	Ergonomist	3 (0.8%)
	Other	54 (14.1%)
Gross domestic product (GDP) per capita	Low	81 (21.1%)
	Middle	117 (30.5%)
	High	186 (48.4%)
Main activity	Practitioner	157 (40.9%)
	Academician	94 (24.5%)
	Researcher	74 (19.3%)
	Other	59 (15.4%)
Working for:	Academia/University	126 (32.8%)
	Governmental/public institution	106 (27.6%)
	Private company	93 (24.2%)
	Self employed	28 (7.3%)
	Non-profit occupational health agency	8 (2.1%)
	Other	23 (6.0%)
Year of joining International Commission on Occupational Health (ICOH)	Before 2010	171 (47.8%)
	2011 and after	187 (52.2%)

**Table 2 ijerph-18-04665-t002:** OSH rules and provisions, training procedures and tools: level of enforcement and utilization.

	Do You Have in Your Country a Set of Rules and Provisions? *	Are There in Your Country Training Procedures and Tools? **
0. No	5 (1.3%)	34 (9.5%)
1. Yes	371 (98.7%)	324 (90.5%)
	**If Yes, is it effectively enforced?**	**If Yes, are they effectively utilized?**
1. Not at all	24 (6.5%)	13 (4.0%)
2. Partially	144 (39.0%)	147 (45.4%)
3. Fairly	87 (23.6%)	102 (31.5%)
4. Much	96 (26.0%)	59 (18.2%)
5. Totally	18 (4.9%)	3 (0.9%)

* *n* = 376; two I don’t know and six missing. ** *n* = 358; 14 I don’t know and 12 missing.

**Table 3 ijerph-18-04665-t003:** Mean value (SD) of the synthetic coverage index unadjusted and adjusted for HDI. Two-way ANCOVA (IV: country e GDP per capita, covariates: HDI).

Country	GDP Per Capita	Unadjusted Synthetic Coverage Index M (SD)	Adjusted * Synthetic Coverage Index M_adj_ (SE)
Africa	Low	0.536 (0.238)	0.849 (0.078)
	Middle	0.579 (0.226)	0.704 (0.056)
	High	--	--
America	Low	0.396 (0.144)	0.552 (0.113)
	Middle	0.630 (0.198)	0.665 (0.024)
	High	0.669 (0.123)	0.535 (0.046)
Asia	Low	0.575 (0.237)	0.772 (0.054)
	Middle	0.694 (0.174)	0.731 (0.043)
	High	0.671 (0.178)	0.556 (0.047)
Europe	Low	0.708 (0.954)	0.765 (0.108)
	Middle	0.639 (0.176)	0.624 (0.044)
	High	0.700 (0.181)	0.574 (0.034)
Oceania	Low	--	--
	Middle	--	--
	High	0.797 (0.093)	0.634 (0.056)

* HDI fixed equal to 0.80390.

**Table 4 ijerph-18-04665-t004:** Sources consulted for professional training in OSH. Multiple choice question.

	Responses		Percentage of Cases
	n	Percentage	
Peer reviewed journals	307	18.3%	81.9%
Websites	307	18.3%	81.9%
Conference proceedings	242	14.4%	64.5%
Monographs and textbooks	212	12.6%	56.5%
Open Access	200	11.9%	53.3%
International grey literature	145	8.6%	38.7%
National grey literature	126	7.5%	33.6%
Not peer reviewed journals	67	4.0%	17.9%
Not open access	41	2.4%	10.9%
Other	35	2.1%	9.3%
Total	1682	100.0%	448.5%

## Data Availability

Data sharing not applicable.

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
