# Peer review of "Contribution of ICOH to Address the Different OSH Needs among Countries: Results of a Survey"

_ijerph, 2021, doi:10.3390/ijerph18094665_

Round 1
Reviewer 1 Report
Thank you for asking me to review the manuscript titled: Contribution of ICOH to address the different OSH needs among countries: Results of a survey
It appears that the purpose of the study was to explore how the level of engagement of the ICOH members contributed to OSH at either a national or international level. OSH (based on the survey) appeared to be defined as rules and provision, trainings and procedures, and tools to improve workers’ awareness and knowledge on the protection of health and safety in the workplace. This study presents the results of a survey of 1929 ICOH members conducted in 2018.
Overall, the authors did not make a strong case for the need for this study and why the study’s findings are relevant or important. I found myself continually asking myself “what were the main research questions that the authors were trying to answer?” and couldn’t work out what they were, or why they had decided to conduct this study. They need to make this clearer.
General comment: This paper has many long and confusing sentences which make reading the manuscript difficult and cumbersome. There are many grammatical errors. There is overuse of nominalizations and passive tense which could be edited to improve readability.
Abstract:
- This sentence is very long, has several grammatical errors and also does not make sense. Please shorten and edit for clarity: In this perspective, the International commission on Occupational Health (ICOH), the oldest and leading global professional organization in the OSH field, developed the present study with the aim of mapping OSH rules and provisions coverage and their level of enforcement at country level worldwide in order to investigate how the level of engagement of the ICOH scientific community has contributed to OSH at national and international level.I would suggest restructuring this sentence to focus on the study’s overall purpose.
- Include the year the survey was conducted.
- It is misleading in the abstract regarding the sample size of the respondents of the survey as they are not included. Please state the respondent sample size and the response rate, rather than the eligible sample. The sample size appears to be 384 (response rate 20%).
- The number of questions in the survey and the survey platform is not the most important information to be including the abstract. Focus on the information that is the most important for the reader to know.
- The abstract should contain the key findings, overall conclusions and implications for environmental research and public health for the study.
Keywords:
The authors have only selected three keywords. Please select the required key words. These should be keywords that would be used by researchers or practitioners who might be looking for research or articles to inform practice. Using MESH terms on key topic areas is a good place to start for selecting keywords.
Introduction:
Overall your introduction could be restructured for better clarity. Ensure that each paragraph has a clear topic. The introduction needs to be concise succinct and just tell the reader the relevant points. As it is written currently, I am unclear what the study is focusing on and what the important and relevant information is. There is too much superfluous information in this section. Sentences can be very long and difficult to follow. Some paragraphs contain just one sentence (e.g. paragraph 2). The importance and need for this study is also not emphasized.
- The first paragraph contains references to agendas and plans that are out dated. There are newer, more recent frameworks, agendas, plans that could be cited here. Although the survey and this study may have been conducted a few years ago now, I would still focus, or at least include, current agendas etc., so that you can position this study as being currently relevant. I think perhaps this is partially covered in Paragraph 4 (Lines 55-60). But this could be restructured to be more succinct and just focus on the pertinent points. Th
- Line 41: “shined a light” should be “shone a light” – to move away from this colloquial phrase you could say “highlighted”
- Your second paragraph is one sentence. I think this is a continuation of the previous (first) paragraph. Please restructure these two paragraphs for clarity.
- I am not sure how paragraph 3 (Lines 47-54) relates to the purpose of this study. It perhaps emphasizes the importance of OSH universally but this is not tied in well.
- The two paragraphs on ICOH can be paired down to a couple of sentences.
- I am not sure what the “Global Survey on occupational Health Services in Selected Secretaries….” is. This whole paragraph is confusing. This should be in the methods section as it is confusing being introduced here. I think what you are trying to say is that, this survey was part of another larger project, right? You need to be clearer. If you want to introduce this here simply state the finding. In a study of 49 ICOH countries, a wide gap in the implementation of OHS policies into practice was found.(citation) This has implications has for xyz. This also reinforces the importance of conducting a further survey of ICOH members on XYZ. Then you can introduce your survey and the purpose of your study.
- There is no clear study purpose statement in the introduction. For example, The purpose/aims of this study was to……..
Methods:
There needs to be an introduction to your methods section.
- I would include a section on “Context” and put the ICOH information and the information on the “Global Surveys etc” that this survey was a part of here.
- There needs to be a statement of the study design. “This was a cross-sectional survey study.”
- How was the pilot of the survey conducting? Was this using some method of cognitive interviewing? Can you provide some brief explanation of the process. What languages was the survey distributed in? How did you ensure interpretability across members who may not have had those languages as their first language?
- Be consistent in use of the terms “subjects” and “participants” throughout for clarity rather than swapping between using the terms.
- “Year of Join to ICOH” - this phrase is not grammatically correct. Is this the phrase that was used in the survey? Although I assume this was understood, can you edit for the paper to be grammatically correct?
- Can you present the variables in the survey in a more reader friendly format. For example, paragraph Lines 139-143 are just a list of variables that appear to be some of the demographic variables but there is not context to how any of the variables are grouped in the methods. Also, within this section there is some explanation of how the analysis relationships are interpreted which makes it even more confusing.
- Many readers, like myself, may not be familiar with this type of analytic approach (i.e. Multiple Correspondence Analysis). Please provide a stand-alone “Statistical Analysis” section in the methods. This information is woven into many different sections across the methods, but there needs to be clear description of what this approach is, how it is performed and interpreted. It would be helpful for the reader to put this in a standalone section as is common in academic papers.
Results:
- Is there a way to present your data in the tables so that the tables don’t take up some much space?
- After reading the results section, I am still very unclear what the aims of this study were. It appears to be a descriptive study. If there were clearer aims presented and a clearer rationale as to the reason and importance for this study, these findings may have more meaning.
Discussion:
- I am not sure that the innovation of the study is the involvement of the whole ICOH active membership, when the response rate was only 20%. You also did not present whether those that responded are representative of the whole membership. If you can provide data to support that these 20% are representative, then you might be able to make this statement. Otherwise, this is actually a limitation of the study.
Author Response
Thank you for your productive inputs and suggestions. We found them useful to improve the manuscript.

Reviewer 2 Report
Thank you for a well-written manuscript. This study identified current status at an overall level and I understand that this is ICOH's role. However, a discussion about how the ICOH can facilitate the development of OSH in the countries that do not have the opportunity to attend the ICOH conferences or have access to scientific articles had been of importance.
Author Response

(The authors gave the same response as above.)

Reviewer 3 Report
Thank you very much for the opportunity to review this paper title: “Contribution of ICOH to address the different OSH needs among countries: Results of Survey”. In its current form, the manuscript is still in a very initial stage, pretty far from the quality standards of an academic paper. So, I suggest a major revision for this paper. I explain some of my reservations in detail below
- Abstract:It is suggested you explain the objective of the research in the abstract. The research methods you use in this study is not clearly explaining in the abstract. Similarly, the results and conclusion of the study are not explaining well in the abstract.
- Too many unknown acronyms are used along with the text (OSH, ICOH, UHC, MCA, HDI, ILO………….). These acronyms affect the article's understanding and finally seem to be annoying. So, I suggest you reduce acronyms.
- Introduction:The introduction is not written well. The introduction should be much more focused. The research objectives should be much more straightforward. Perhaps it could be helpful to articulate the research question explicitly. Similarly, authors need to clearly state the paper's value-added and better discuss how this work could be worth it for both academics and practitioners. I suggest you explain the answer to mentioned four questions in your revised introduction part. Moreover, I have suggested below mentioned paper it will help you to improve your introduction.
- What are the academic research questions of this study?
- I suggest to the authors in the last paragraph of the introduction explains the structure of the paper.
- I also suggest you write your research background as a story
- Wang, Z., Zaman, S., Rasool, S. F., uz Zaman, Q., & Amin, A. (2020). Exploring the Relationships Between a Toxic Workplace Environment, Workplace Stress, and Project Success with the Moderating Effect of Organizational Support: Empirical Evidence from Pakistan. Risk Management and Healthcare Policy, 13, 1055.
- Literature Revie:Arguments do not flow logically, and ideas are not well connected. It is difficult to figure out the research strategy followed. So, I suggest you create a new heading with the title of “Literature review”. Under this heading, explain occupational safety and health, International Commission on Occupational Health and explain how OSH rules and provisions coverage and their level of enforcement at country level world-18 wide. Moreover, how the level of engagement of the ICOH scientific community has 19 contributed to OSH at the national and international level.?.
- I suggest you explain how these all concepts are integrated and how previous studies support your study.
- What theory supports your study? It is suggested to explain the theory which supports your finding, and the reflation of this theory must show throughout the manuscript.
- Some paragraphs need to be read twice in order to understand what the authors are trying to communicate. Use more straightforward language. Paragraphs are extremely long.
- It is suggested to the authors provide the comprehensive research model of your study without model, this study does not actually make much sense. (below mentioned paper will help you what kind of research model I want to see in your study).
- Finally, can you state the core elements in your research that bring novelty to previous literature?
- Rasool SF, Wang M, Tang M, Saeed A, Iqbal J. How Toxic Workplace Environment Effects the Employee Engagement: The Mediating Role of Organizational Support and Employee Wellbeing. International Journal of Environmental Research and Public Health. 2021; 18(5):2294. https://doi.org/10.3390/ijerph18052294
- Research Methodology: The research methodology of this study is not explained well.
- Which research approach the authors used in this study? (Develop sub-heading with the title of Research approach and explain what research approach you used in this study and why you use this research approach specifically in your study.)
- Which sampling techniques used for this study?
- It is also suggesting that the authors provide the details about the instrument's validity and how authors pre-test the instrument?
- Explain the “Sample description” in the research method instead of the results section.
- I suggested that once you carefully read the research methodology of the below mentioned study, it will help you improve the study.
- Rasool, S.F.; Maqbool, R.; Samma, M.; Zhao, Y.; Anjum, A. Positioning Depression as a Critical Factor in Creating a Toxic Workplace Environment for Diminishing Worker Productivity. Sustainability 2019, 11,2589. https://doi.org/10.3390/su11092589
- Results: The results and analysis are fine.
- Discussion: This section is feeble; it is not well explained. I suggest explaining the discussion part in detail and integrate it with the results and previous literature.
- Conclusion or Policy Implication: The conclusion or policy implication is required to explain in this study. I suggest you explain the novel contribution of your study to occupational safety and health. I also suggested you explain what policies can be improving the safety and health among different counties or in the organizations. Furthermore, the conclusion or policy implication must be integrating with the introduction, theory, and your findings.
- References: It is recommended to make use of recent references to support these arguments (ideally, published during the past 5 years).
Author Response

(The authors gave the same response as above.)

Round 2
Reviewer 3 Report
In its current form, the manuscript is up to the standards of an academic paper. So, I accept this paper.